# ABO blood group and the risk of aortic disease: a nationwide cohort study

Igor Zindovic,[1] Gustaf Edgren,[2,3] Shahab Nozohoor ,[1] Ammar Majeed[2,4]

► Prepublication history and supplemental material for this paper are available online. To view these files, please visit the journal online (http://dx.doi.org/10.1136/bmjopen-2019-036040).

[1]Department of Clinical Sciences, Department of Cardiothoracic Surgery, Lund, Sweden, Lund University, Skåne University Hospital, Lund, Sweden
[2]Department of Medicine, Solna, Clinical Epidemiology Division, Karolinska Institutet, Stockholm, Sweden
[3]Department of Cardiology, Södersjukhuset, Stockholm, Sweden
[4]Department of Gastroenterology, The Alfred Health, Melbourne, Victoria, Australia

**Correspondence to**
Dr Igor Zindovic;
igor.zindovic@med.lu.se

## ABSTRACT

**Objectives** To analyse the association between ABO blood group and aortic disease using data on blood donors and transfused patients from Sweden.

**Design** This was a retrospective study using data from the Swedish portion of the Scandinavian Donations and Transfusions Database. The association between ABO blood group and aortic disease was analysed using log-linear Poisson regression models and presented as incidence rate ratios (IRRs).

**Setting** Swedish population-based study.

**Participants** The study cohort consisted of 1 164 561 Swedish blood donors and 961 637 transfused patients with a combined follow-up time of 29 390 649 person-years.

**Primary and secondary outcome measures** IRRs of aortic events (ie, aortic aneurysms and/or aortic dissections) in relation to patient blood group.

**Results** A total of 20 684 aortic events occurred during the study period. Non-O donors and patients had similar incidence of aortic events when compared with blood group O donors and patients with an IRR of 0.98 (95% CI, 0.93–1.04) and 1.00 (95% CI, 0.97–1.03), respectively. There were no differences between non-O and blood group O individuals when aortic dissections and aortic aneurysms were analysed separately. Blood group B conferred a lower risk of aortic aneurysms in the patient cohort when compared with blood group O (IRR, 0.90; 95% CI, 0.85–0.96).

**Conclusions** In the present study, there were no statistically significant associations between ABO blood group and the risk of aortic disease. A possible protective effect of blood group B was observed in the patient cohort but this finding requires further investigation.

## INTRODUCTION

Aortic aneurysm and dissection are potentially life-threatening conditions that share common risk factors—age, hypertension, atherosclerosis and smoking—with other types of cardiovascular disease.[1–3] The association between ABO blood groups and cardiovascular disease and venous thromboembolism (VTE) has been widely reported, but the pathophysiological mechanisms are not fully understood.[4–8]

Few studies have examined the association between ABO blood group and aortic disease.[9–11] In an analysis of 504 patients with abdominal aortic aneurysm (AAA), Viklander

---

### Strengths and limitations of this study

► Data were collected from a quality registry covering all blood donations and transfusions 1968–2012.
► This study is strengthened by complete follow-up of the study cohort.
► This is by far the largest study assessing the association between blood groups and aortic disease.
► The study is limited by donors representing a selected cohort of healthy individuals, whereas the patient cohort has a higher prevalence of comorbidities compared with the general population.
► Another limitation is that detailed information on comorbidities of individuals in the study population was not known.

---

*et al* did not find a significant difference in distribution of blood groups between patients with AAA or ruptured AAA compared with a population-based control group.[9] Fatic *et al* on the other hand, demonstrated that a significantly higher proportion of patients with AAA had blood group O compared with a control group consisting of blood donors from the general population of Montenegro.[10] Available research is limited by the small size of the studies, conflicting results and the omission of patients with aortic dissection, warranting further examination of the relationship between blood groups and aortic disease.

The aim of the present study was to evaluate the association between ABO blood group and aortic disease using data from a large cohort of Swedish individuals derived from the Scandinavian Donations and Transfusions (SCANDAT2) Database.[12]

## MATERIAL AND METHODS
### Data source

The study was based on the SCANDAT2 Database. This is a computerised, combined donation and transfusion register using data on more than 1.6 million blood donors and 2 million recipients of blood products in Sweden from 1968 to 2012 and in Denmark from 1981 to 2012.[12 13] Using the unique

BMJ

personal identity numbers assigned to all residents of the two countries, SCANDAT2 is linked to national population, death and migration registers, allowing for the complete follow-up of all individuals in the register up to and including 31 December 2012.[14] The database is also linked to the national inpatient, outpatient, cause of death and cancer registries with information on registered diagnoses coded as the International Classification of Diseases (ICD) codes, versions 6–10.

## Study design

This was a retrospective data analysis based on individuals included in the SCANDAT2 Register. For the purpose of these analyses, we only used data from the Swedish portion of the SCANDAT2 Database and restricted the analysis to donors and patients who donated or received their first blood product after 1 January 1970, when the patient register in Sweden had good national coverage. Donors and recipients with a documented blood group in SCANDAT2, who had given or received at least one blood product between 1 January 1970 and 31 December 2012, were included in the study. We used a set of ICD codes to identify donors and recipients who were diagnosed with either aortic aneurysm or aortic dissection (online supplemental table 1) based on information from the patient register. The different ICD codes were grouped into two different disease groups based on aetiological factors or disease characteristics (online supplemental table 2). Recognising that aortic aneurysm and dissection share common risk factors and disease characteristics, we created a third group that encompassed all ICD codes for aortic aneurysm and dissection. Donors and recipients who had a diagnosis of aortic aneurysm and/or dissection prior to their first donation or transfusion were excluded from the study. In the present study, we used the term 'aortic disease' to reflect either aortic aneurysm or aortic dissection and the diagnosis of an aortic aneurysm or aortic dissection as an 'aortic event'.

## Statistical analysis

Recognising differences in demographics and comorbidities, we created separate cohorts for the donors and the recipients. In the analysis for the donor cohort, we followed the donors from the date of first electronically recorded blood donation in SCANDAT2 or the study start date (1 January 1970), whichever came last. For the transfusion recipient cohort, individuals were followed from the start of the study (1 January 1970) or 365 days after their first electronically recorded blood transfusion in SCANDAT2, whichever came last. We delayed the patients' entry into the study by 1 year after their first transfusion to reduce the risk of reverse causation bias (ie, when the indication for their first transfusion would somehow be related to the subsequent occurrence of an aortic event). All study participants were followed up until either the date of diagnosis of aortic aneurysm or dissection, emigration, death or end of follow-up (31 December 2012), whichever came first. As an individual could be diagnosed with more than one type of aortic aneurysm or have an aneurysm and dissection diagnosed at different time points during their lifetime, we allowed for the same individual to contribute time at risk and events into more than one disease group based on their diagnoses.

Baseline characteristics are presented as means and SD for continuous variables, and frequencies and percentages for categorical ones. The ratio of the incidence of an aortic aneurysm or dissection in individuals with blood group A, AB or B compared with those with blood group O (the incidence rate ratio (IRR)) was analysed using log-linear Poisson regression models. The models included ABO blood group (A, AB, B or O) and were adjusted for age (expressed as natural cubic spline with five equally spaced knots), sex (female or male) and calendar year of observation (expressed as natural cubic spline with five equally spaced knots). The models thus estimated the number of events (of each type) as a function of patient blood group (either parameterised as A, AB, B or O, or as O vs non-O), patient sex, patient age and calendar period of observation, with the logarithm of time at risk as an offset term (online supplemental table 3). In alternative analyses, ABO blood groups were instead classified as O or non-O. Interactions between age and blood group and age and sex were also assessed by including interaction terms in the Poisson regression model.

All data processing and statistical analyses were performed with SAS statistical analysis software (V.9.4). All p values below 0.05 were considered statistically significant.

## Patient and public involvement

Apart from contributing with their personal study data, patients were not involved in this study.

# RESULTS

## Study population and follow-up

The donor cohort consisted of 1 164 561 blood donors with a median follow-up time of 17.9 years, and the patient cohort included 961 637 recipients of blood transfusions with a median follow-up time of 8.9 years. Donors had a median age of 28 years at entry and were less often women (47.1%), whereas patients had a median age of 69 years, and were predominantly women (58.3%). The distribution of the different blood groups was similar in the two cohorts, with blood group O comprising about 38% of each cohort (table 1).

## Overall risk of aortic event

Altogether, there were 716 116 donors with a non-O blood group and 448 445 with blood group O, with a total follow-up time of 12 827 612 and 7 990 859 person-years, respectively. We observed 3086 events (aortic aneurysm or dissection) in donors with non-O blood group compared with 1977 events in donors with blood group O. The resulting IRR for any aortic event in donors with a non-O blood group compared with blood group O was

| Table 1 | Characteristics of study participants | |
|---|---|---|
| | **Donor cohort** | **Patient cohort** |
| Number of subjects | 1 164 561 | 961 637 |
| Female sex, N (%) | 548 234 (47.1) | 560 537 (58.3) |
| Age at start of follow-up, N (%) | | |
| 18–40 years | 910 537 (78.2) | 134 160 (14.0) |
| 41–60 years | 246 772 (21.2) | 185 080 (19.2) |
| 61–75 years | 7252 (0.6) | 307 216 (31.9) |
| >75 years | 0 (0.0) | 335 181 (34.9) |
| Median (IQR) | 28 (22–38) | 69 (54–79) |
| ABO blood group, N (%) | | |
| A | 522 364 (44.9) | 444 075 (46.2) |
| AB | 62 001 (5.3) | 47 610 (5.0) |
| B | 131 751 (11.3) | 101 355 (10.5) |
| O | 448 445 (38.5) | 368 597 (38.3) |
| Born in Sweden, N (%) | 1 076 286 (92.4) | 858 098 (89.2) |

0.98 (95% CI, 0.93–1.04) (table 2). The patient cohort included 593 040 patients with a non-O blood group and 368 597 patients with blood group O with a combined follow-up time of 5 266 210 and 3 305 968 person-years, respectively. Throughout follow-up, an aortic event (aortic aneurysm or dissection) was diagnosed in 9567 patients with a non-O blood compared with 6054 events in those with blood group O. The IRR for such events in patients with non-O blood was 1.00 (95% CI, 0.97–1.03) compared with those with blood group O.

No significant differences were observed when blood groups A, B, AB were compared with blood group O except for blood group B having an IRR 0.90 (95% CI, 0.85–0.96) for the occurrence of any aortic event in the patient cohort (tables 2–4).

### Risk of aortic dissection

Aortic dissection occurred in 587 non-O donors and 414 O donors during the study period, corresponding to an IRR of 0.89 (95% CI, 0.78–1.01) for non-O donors compared with blood group O donors (table 3). Aortic dissection was observed in 1272 patients with non-O blood and 799 patients with blood group O, with an IRR

of 1.01 (95% CI, 0.92–1.10). There were no differences in risk of aortic dissection in patients with blood group A, B or AB compared with those with blood group O (table 3).

### Risk of aortic aneurysm

Aortic aneurysm was diagnosed in 2781 donors with non-O blood and 1740 O donors, resulting in an IRR of 1.00 (95% CI, 0.95–1.07) for non-O donors compared with O donors (table 4). For the patient cohort, the corresponding numbers were 8906 and 5648, respectively, resulting in an IRR of 1.00 (95% CI, 0.96–1.03). Similar results were obtained in the analysis of the risk of aortic aneurysm in patients with blood groups A and AB. However, patients with blood group B had a significantly lower risk of experiencing an aortic event when compared with patients with blood group O with an IRR of 0.89 (95% CI, 0.84–0.94) (table 2).

### DISCUSSION

Here, we present results from the hitherto largest study of the association between ABO blood groups and aortic aneurysm or dissection with more than 2 million Swedish blood donors and transfusion recipients, a total of 20 684 aortic events and a combined follow-up time of almost 30 million person-years. The present study did not demonstrate a statistically significant association between ABO blood group and aortic disease with the exception of an inconsistent association in the patient cohort between blood group B and aortic aneurysm.

Previously, Fatic et al showed that a higher proportion of patients diagnosed with AAA had blood group O when compared with blood donors in the normal population,[11] whereas Viklander et al[9] could not find any association between blood groups and the risk of developing AAA or AAA rupture. It has been reported that non-O blood groups (ie, blood groups A, B and AB) are associated with significantly higher risk of VTE and cardiovascular disease, including myocardial infarction, peripheral vascular disease and cerebrovascular stroke.[4–8] This could, in part, be explained by blood group O patients having 25%–30% lower levels of circulating Factor VIII (FVIII) and von Willebrand factor (VWF).[15] However, Dentali et

**Table 2** Relative risk of aortic dissection or aortic aneurysm in relation to ABO blood group, presented separately for the donor and recipient cohorts

| ABO blood group | Donor cohort | | Patient cohort | |
|---|---|---|---|---|
| | Events/person-years | Incidence rate ratio (95% CI) | Events/person-years | Incidence rate ratio (95% CI) |
| A | 2281/9 344 274 | 1.00 (0.94 to 1.06) | 7343/3 911 097 | 1.02 (0.98 to 1.05) |
| AB | 268/1 125 603 | 0.94 (0.83 to 1.07) | 783/426 761 | 1.04 (0.96 to 1.12) |
| B | 537/2 348 996 | 0.93 (0.85 to 1.03) | 1441/907 053 | 0.90 (0.85 to 0.96) |
| O | 1977/7 984 924 | 1.00 (ref) | 6054/3 292 052 | 1.00 (ref) |
| Non-O | 3086/12 818 872 | 0.98 (0.93 to 1.04) | 9567/5 244 912 | 1.00 (0.97 to 1.03) |
| O | 1977/7 984 924 | 1.00 (ref) | 6054/3 292 052 | 1.00 (ref) |

**Table 3** Relative risk of aortic dissection in relation to ABO blood group, presented separately for the donor and recipient cohorts

| ABO blood group | Donor cohort | | Patient cohort | |
|---|---|---|---|---|
| | Events/person-years | Incidence rate ratio (95% CI) | Events/person-years | Incidence rate ratio (95% CI) |
| A | 426/9 350 727 | 0.89 (0.77 to 1.01) | 933/3 927 796 | 0.98 (0.89 to 1.08) |
| AB | 55/1 126 318 | 0.93 (0.70 to 1.23) | 116/428 385 | 1.16 (0.95 to 1.41) |
| B | 106/2 350 567 | 0.88 (0.71 to 1.09) | 223/910 029 | 1.05 (0.91 to 1.22) |
| O | 414/7 990 859 | 1.00 (ref) | 799/3 305 968 | 1.00 (ref) |
| Non-O | 587/12 827 612 | 0.89 (0.78 to 1.01) | 1272/5 266 210 | 1.01 (0.92 to 1.10) |
| O | 414/7 990 859 | 1.00 (ref) | 799/3 305 968 | 1.00 (ref) |

al[6] demonstrated that non-O patients were at an increased risk of VTE even after adjustment for different levels of FVIII/VWF. Furthermore, it has been shown that the genetic variation of ABO is associated with atherosclerosis and biomarkers of vascular inflammation, both of which are recognised risk factors for aortic aneurysm formation and aortic dissection.[1 16 17] The normal physiology of the vascular system relies on a negatively charged vascular wall, which repels the similarly negatively charged red blood cells and many negatively charged circulating proteins. A reduction of the negative charge may make the vascular wall more prone to thrombus formation and may also have an effect on the transportation of charged circulating proteases and cytokines into the vessel wall.[18 19] ABO blood groups and Rhesus antigen status have both been shown to influence the size of the charge potential,[20] but the findings of the present study indicate that the previously described pathophysiological mechanisms of cardiovascular disease associated with ABO blood groups do not seem to translate to the formation of aortic aneurysms or the development of aortic dissection.

The differences in risk of cardiovascular disease between non-O and O individuals have previously been suggested as a specific effect of blood group A.[21 22] The analyses of different ABO blood groups in the present study could not demonstrate a specific effect of blood group A. Instead, our analyses showed a protective effect of blood group B on aortic disease in the patient cohort, primarily driven by significantly lower risk of aortic

aneurysm development. To our knowledge, there are no previous reports suggesting an association between blood group B and cardiovascular disease and, thus, we have no plausible pathophysiological explanation for the specific effect of blood group B in this study. Furthermore, this association was not demonstrated in the donor cohort and, therefore, our speculation is that the protective effect of blood group B on the incidence of aortic disease might have been a chance finding.

The results presented in this study need some careful consideration. First, the donor and the patient cohort both represent selected populations and not the general population. However, to our knowledge, there are no data on blood group distributions in large cohorts of randomly selected individuals and therefore, donors or patients have generally served as control groups in similar studies in the past.[4–9 12 13] Donors represent a selected cohort of healthy individuals, whereas the patient cohort has a higher prevalence of co-morbidities compared with the general population. Despite the differences in health profile, this study did not demonstrate any major differences in blood group distribution between these two groups and therefore, we have no reason to believe that the distributions in donors and patients are not reflective of those in the general population. The external validity of the study might, therefore, be affected. Aortic aneurysm and dissection generally affect individuals older than the average age of blood donors. However, we did not find a significant interaction between ABO blood group and age. Furthermore, The SCANDAT2 Database does not

**Table 4** Relative risk of aortic aneurysm in relation to ABO blood group, presented separately for the donor and recipient cohorts

| ABO blood group | Donor cohort | | Patient cohort | |
|---|---|---|---|---|
| | Events/person-years | Incidence rate ratio (95% CI) | Events/person-years | Incidence rate ratio (95% CI) |
| A | 2056/9 345 277 | 1.02 (0.96 to 1.09) | 6852/3 912 179 | 1.02 (0.98 to 1.05) |
| AB | 242/1 125 746 | 0.97 (0.85 to 1.11) | 733/426 928 | 1.04 (0.97 to 1.13) |
| B | 483/2 349 202 | 0.95 (0.86 to 1.05) | 1321/907 322 | 0.89 (0.84 to 0.94) |
| O | 1740/7 985 856 | 1.00 (ref) | 5648/3 292 796 | 1.00 (ref) |
| Non-O | 2781/12 820 224 | 1.00 (0.95 to 1.07) | 8906/5 246 429 | 1.00 (0.96 to 1.03) |
| O | 1740/7 985 856 | 1.00 (ref) | 5648/3 292 796 | 1.00 (ref) |

include patient genotyping and, therefore, we cannot fully assess the distribution of O alleles in blood group A and B patients. The use of hospital registers carries the inherent risk of misclassification, but it is unlikely that it relates to patient blood group. The limitations of the ICD diagnosis classification system did not permit us to definitively separate non-ruptured from ruptured aortic aneurysms and, thus, these were analysed together, despite ruptured aortic aneurysms potentially having larger similarities with aortic dissections. This should not, however, influence results from the analyses of the combination of aortic aneurysm and dissection.

On the other hand, there are several strengths to this study. It is by far the largest study investigating the association between ABO blood groups and aortic disease. The use of nationwide population and health registers allowed for complete follow-up and adds to the strength of the study. Blood group data were recorded prospectively and have high validity. Even though patients are potentially aware of their blood group, it is unlikely that such awareness has influenced their behaviour. The prevalence of other risk factors for aortic disease should be similar between the different blood groups and, thereby, it should not have a confounding effect on the results of the analyses.

In conclusion, this large study did not demonstrate a significant association between ABO blood group and the risk of aortic disease. A possible protective effect of blood group B was, however observed in the patient cohort. Further studies are needed to confirm the presence and strength of this association.

**Contributors** The conception and design of the study were performed by IZ, GE and AM. GE and AM performed the statistical analyses; and IZ and SN produced the first draft of the manuscript. The data were interpreted by all coauthors. The manuscript was critically reviewed for important intellectual content and the final version was approved by all authors. All authors had full access to all the data (including statistical reports and tables) of the study and can take responsibility for the integrity of the data and the accuracy of the data analysis.

**Funding** The SCANDAT2 Database was made possible through grants to Gustaf Edgren from the Swedish Research Council (2017-01954) and the Danish Research Council.

**Competing interests** None declared.

**Patient consent for publication** Not required.

**Ethics approval** The creation of SCANDAT2 and the conduct of this study were approved by the Swedish Ethical Review Authority, the Danish Scientific Ethics Committee and the Danish Data Protection Agency.

**Provenance and peer review** Not commissioned; externally peer reviewed.

**Data availability statement** No additional data are available.

**ORCID iD**
Shahab Nozohoor http://orcid.org/0000-0002-5786-864X

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
