## [Reviewer comments · BMJ Open]

ARTICLE DETAILS

TITLE (PROVISIONAL)	ABO blood group and the risk of aortic disease - A nationwide cohort study
AUTHORS	Zindovic, Igor; Edgren, Gustaf; Nozohoor, Shahab; Majeed, Ammar

VERSION 1 – REVIEW

REVIEWER	Yuk Law The University of Hong Kong
REVIEW RETURNED	23-Jan-2020

GENERAL COMMENTS	Thank you for the invitation to review this article to see if ABO blood group has any relationship to aortic diseases. The article is well written and findings were clearly presented. I has two questions: (1) Methodology: Could the authors explain why using person-years as the denominator for calculation of risk (rather than just simple prevalence)? A person was born with the specific blood group and the blood group was found out (not the onset of disease) after their first blood donation or transfusion. Should the beginning of calculation start from the date of birth rather than the date of registration? (2) Methodology: I agree that patient with aortic aneurysm can have a separate dissection later. However, patient with aortic dissection usually develops aneurysm later as a natural progression and hence should be calculated just once rather than two events. Is there any audit on this respect?
---

REVIEWER	Dr Ashutosh Hardikar Menziess institute of Research, University of Tasmania, Hobart and Royal Hobart Hospital, Hobart, Tasmania, Australia
REVIEW RETURNED	17-Feb-2020

GENERAL COMMENTS	congratulations on this very large study which clarifies the debate around ABO blood groups and aortic disease. The objective was clear and the results well presented. The conclusions are appropriate and important and I agree that the protection offered by blood group B in the recipients group against aortic aneurysm seems to be a chance finding. I have 2 questions: 1] Why did you choose Poisson regression model for analysis over other regression methods? I believe the Poisson regression model assumes that the mean and variance are the same, I could not see the mean or variance clearly stated in any of the tables or supplementary materials. 2] The only adjustment that has been done in your model is for age,
--

	gender and the calendar year. Could you elaborate how other confounding factors which might obviously affect the occurrence of aortic disease (both aneurysm or dissection) were accounted for? One could say that significant aortic valve disease, systemic hypertension and syndromic disorders like Marfan's syndrome would affect the occurrence of aortic disease much more and hence the data should be adjusted to these factors as well. Thank you once again on this extensive study and look forward to your response.
--	--

REVIEWER	Anup Karan Indian institute of Public Health - Delhi, Public Health Foundation of India
REVIEW RETURNED	06-Apr-2020

GENERAL COMMENTS	I was asked to conduct only statistical review of the paper. I find the statistics used in this study is right. Authors have used log-linear Poisson regression model to estimate incidence rate ratio of an aortic aneurysm or dissection. The statistical techniques looks fine to me although I would like to see the exact regression specification (equation). Such model can be presented as a supplementary material for a greater clarity to readers. Yest another technical question, I have in mind is how did authors addressed any co-morbidity situation either in donors or recipient.
---

VERSION 1 – AUTHOR RESPONSE

Reviewer 1

Comment 1.

Methodology: Could the authors explain why using person-years as the denominator for calculation of risk (rather than just simple prevalence)? A person was born with the specific blood group and the blood group was found out (not the onset of disease) after their first blood donation or transfusion. Should the beginning of calculation start from the date of birth rather than the date of registration?

Response: We thank the reviewer for this question. The individuals in SCANDAT had different follow-up times, with some individuals contributing with shorter follow up times than others, as individuals are censored when they develop an aortic aneurysm/dissection, at the time of migration, death or end of study period. What the reviewer is suggesting, i.e. to compute instead incidence proportions (i.e. proportion diseased during follow-up), would thus be less attractive and instead we have using person-time as the denominator, allowing us to compute incidence rates. The resulting measures of association would more accurately reflect the rate of development of aortic aneurysm/dissection in our population. Furthermore, the entry point into the study was taken to be the date of donation/transfusion (rather than birthdate)is used to avoid possible effects of survival bias, i.e. where our estimates would be biased from non-survival of individuals diagnosed before the actual start of follow-up.

Comment 2.

Methodology: I agree that patient with aortic aneurysm can have a separate dissection later. However, patient with aortic dissection usually develops aneurysm later as a natural progression and hence should be calculated just once rather than two events. Is there any audit on this respect?

Response: We agree with the reviewer and as suggested by the reviewer, the study subjects were only followed up until their first event in the study database, as stated in the Material and Methods section: "All study participants were followed up until either the date of diagnosis of aortic aneurysm or dissection, emigration, death or end of follow-up (December 31, 2012), whichever came first." As a consequence, follow-up was terminated without regard to if the patient had a post dissection aneurysm or not.

Reviewer 2

Comment 1.

Why did you choose Poisson regression model for analysis over other regression methods? I believe the Poisson regression model assumes that the mean and variance are the same, I could not see the mean or variance clearly stated in any of the tables or supplementary materials.

Response: It is correct that the Poisson model is quite simplistic in that it only estimates one parameter for the equal variance and mean, however it is our experience from conducting these types of incidence calculations over several decades that it is often remarkably accurate and seldom results in overdispersion (which is what one would worry about when the model variation is lesser than the actual variation in the data). As a simple test of whether there was evidence of overdispersion in our data, and whether such overdispersion affected the results, we repeated the analyses using a negative binomial model, which estimates separate parameters for the mean and variance. The negative binomial model yielded identical results. As such, this is as expected not a problem. As this is a niche issue, which in practical terms is non-informative for readers, we have not expanded information about this in the manuscript.

Comment 2:

The only adjustment that has been done in your model is for age, gender and the calendar year. Could you elaborate how other confounding factors which might obviously affect the occurrence of aortic disease (both aneurysm or dissection) were accounted for? One could say that significant aortic valve disease, systemic hypertension and syndromic disorders like Marfan's syndrome would affect the occurrence of aortic disease much more and hence the data should be adjusted to these factors as well.

Response: We acknowledge the reviewer's comment. Firstly, we presented separate results for donors, who are generally a healthy population with limited co-morbidities. The results from this population was comparable to that of the patients' cohort, indicating that additional co-morbidities in the patients' cohort did not influence aortic event rates between the different ABO groups. Secondly, given the lack of association between ABO blood group and aortic valve disease/systemic hypertension/syndromic disorders, and because of the large sample size of the donors and patients cohort, the distribution of such confounding factors can be assumed to be comparable between the different blood groups. As a consequence of that, the influence of other possible confounding factors that were not included in the regression model, on the outcome (rate of aortic dissection/aneurysm between the different blood groups) can be assumed to be minimal, if any.

Reviewer 3

Comment 1.

I find the statistics used in this study is right. Authors have used log-linear Poisson regression model to estimate incidence rate ratio of an aortic aneurysm or dissection. The statistical techniques looks fine to me although I would like to see the exact regression specification (equation). Such model can be presented as a supplementary material for a greater clarity to readers.

Response: We thank the reviewer for this comment and have included a supplementary table with the regression model (equation) parameters and estimates.

Comment 2.

Yes another technical question, I have in mind is how did authors addressed any co-morbidity situation either in donors or recipient.

Response: In principle, we did not account for co-morbidity at all as such factors are very unlikely to be related to ABO blood type on account of the latter being a genetic trait. Still, because we presented separate results for donors, who are generally a healthy population with limited co-morbidities. The results from this population was comparable to that of the patients' cohort, indicating that additional co-morbidities in the patients' cohort did not influence aortic event rates between the different ABO groups. Secondly, given the lack of association between ABO blood group and aortic valve disease/systemic hypertension/syndromic disorders, and because of the large sample size of the donors and patients cohort, the distribution of such confounding factors can be assumed to be comparable between the different blood groups. As a consequence of that, the influence of other possible confounding factors that were not included in the regression model, on the outcome (rate of aortic events between the different blood groups) can be assumed to be minimal, if any.

VERSION 2 – REVIEW

REVIEWER	Yuk Law The University of Hong Kong
REVIEW RETURNED	25-May-2020

GENERAL COMMENTS	The reviewer completed the checklist but made no further comments.
--

REVIEWER	Anup Karan Indian Institute of Public Health Delhi, Public Health Foundation of India, India
REVIEW RETURNED	26-May-2020

GENERAL COMMENTS	In my last review I had recommended to present details of specification of the Poisson regression model in supplementary material. I don't see any mention of this in the manuscript.
---

VERSION 2 – AUTHOR RESPONSE

Reviewer 3 comment:

In my last review I had recommended to present details of specification of the Poisson regression model in supplementary material. I don't see any mention of this in the manuscript.

Response:

Apart from adding Supplementary table 3 to the Supplementary file, we have now added the following information to the Material and Methods section: "The models thus estimated the number of events (of each type) as a function of patient blood group (either parameterized as A, AB, B, or O, or as O vs non-O), patient sex, patient age and calendar period of observation, with the logarithm of time at risk as an offset term (Supplementary Table 3)."